# CVTrack: Combined Convolutional Neural Network and Vision Transformer Fusion Model for Visual Tracking

**DOI:** 10.3390/s24010274

**Published:** 2024-01-03

**Authors:** Jian Wang, Yueming Song, Ce Song, Haonan Tian, Shuai Zhang, Jinghui Sun

**Affiliations:** 1Changchun Institute of Optics, Fine Mechanics and Physics, Chinese Academy of Sciences, Changchun 130033, China; wangjian213@mails.ucas.ac.cn (J.W.);; 2University of Chinese Academy of Sciences, Beijing 101408, China

**Keywords:** single-object tracking, vision transformer, Siamese network, feature fusion

## Abstract

Most single-object trackers currently employ either a convolutional neural network (CNN) or a vision transformer as the backbone for object tracking. In CNNs, convolutional operations excel at extracting local features but struggle to capture global representations. On the other hand, vision transformers utilize cascaded self-attention modules to capture long-range feature dependencies but may overlook local feature details. To address these limitations, we propose a target-tracking algorithm called CVTrack, which leverages a parallel dual-branch backbone network combining CNN and Transformer for feature extraction and fusion. Firstly, CVTrack utilizes a parallel dual-branch feature extraction network with CNN and transformer branches to extract local and global features from the input image. Through bidirectional information interaction channels, the local features from the CNN branch and the global features from the transformer branch are able to interact and fuse information effectively. Secondly, deep cross-correlation operations and transformer-based methods are employed to fuse the template and search region features, enabling comprehensive interaction between them. Subsequently, the fused features are fed into the prediction module to accomplish the object-tracking task. Our tracker achieves state-of-the-art performance on five benchmark datasets while maintaining real-time execution speed. Finally, we conduct ablation studies to demonstrate the efficacy of each module in the parallel dual-branch feature extraction backbone network.

## 1. Introduction

Visual object tracking has been an important topic and research hotspot in the computer vision area, aiming to accurately predict the position of an object in each frame of a continuous video image sequence by identifying the target’s features such as size, shape, color or other characteristics under the condition of only given the initial state of the target. The target tracking task does not have a specified category for the tracking target, and the tracking scene is complex and varied, with problems such as target scale change, target occlusion, motion blur, target disappearance, and so on. Therefore, real-time, accurate, and robust tracking of the target is a highly challenging task.

Single object tracking generally gives a tracking frame in the first frame, and the object in the frame is the one that needs to be tracked by the algorithm in the subsequent frames and the frame in the first frame is usually provided by the detection algorithm or labeled manually. Early object tracking used many classical machine learning methods, such as optical flow, particle filtering, mean shift, etc. Although these algorithms have some limitations, they laid the foundation for the target tracking field to flourish. Correlation filtering methods have had a profound impact on the field of target tracking. Tracking algorithms based on correlation filtering, such as MOSSE [1], CSK [2], KCF [3], BACF [4], SAMF [5], and other algorithms, have transformed the field of communication into correlation filtering (measuring the similarity of two signals) is introduced into target tracking. By using fast Fourier transform (FFT) to speed up the calculation of image similarity, using fast Fourier transform, the tracking frame rate of MOSSE [1] can reach 600 fps, laying the foundation for the real-time application of related filtering series methods. As the application of deep learning becomes more and more widespread, many researchers have successfully introduced deep learning into target tracking, resulting in significant breakthroughs in tracking performance. The early application method of deep learning in the field of target tracking is to directly apply the features learned by the network to the tracking framework of correlation filtering, thereby obtaining better tracking results. As the SiamFC [6] algorithm introduces the twin network into the field of target tracking, the target-tracking algorithm based on the twin network gradually becomes mainstream. It describes target tracking as a matching problem between the template and the search area. The target template area and the search area are extracted through the twin network. Searching for the characteristics of the area and calculating the similarity between them, it achieves a balance between tracking speed and accuracy due to its unique network structure. In recent years, with the introduction of the transformer [7] into the field of target tracking, single-target tracking has once again experienced rapid development. The structure of the transformer is good at capturing the long-range dependence of features and is very suitable for pairwise matching tasks. In addition, the target-tracking algorithm based on a single-stream transformer inputs the target template and search area into a transformer-based backbone network and completes feature extraction and fusion in the same backbone network at the same time. This simple and efficient method further improves the accuracy of target tracking, and its performance is far ahead on multiple large-scale datasets.

However, in practical applications, existing single-object trackers still have limited interpretability for factors such as target deformations, occlusions, field of view, interference from similar objects, and size changes. Designing efficient and robust trackers remains a challenging and meaningful task. The current mainstream target-tracking algorithms use CNN or a transformer as the backbone. CNN is a feature extraction network structure based on convolutional layers, which have the characteristics of local connections, weight sharing, etc. It can extract spatial features in images, such as edges, corners, etc. Adding a pooling layer after the convolutional layer can further reduce the size of the feature map and reduce computational complexity. CNN can also achieve classification and recognition of complex images through a combination of multi-layer convolutional layers and fully connected layers. The transformer is a feature extraction network structure based on a self-attention mechanism, which can perform attention calculations on each position in the input sequence to obtain global contextual information.

Although the transformer model has improved the performance of trackers to some extent, there are still issues in feature extraction. In the process of object tracking, both global modeling and local modeling of target features significantly influence object matching and localization. Through analysis, we found that existing trackers mainly rely on either separate transformer features or CNN features. CNN can extract local information from images, while transformers can capture global information over long distances. In the current research work, two methods are used separately, ignoring the importance of complementary advantages between local and global features. Both local and global features play an indispensable role in improving target-tracking performance. However, how to better combine local and global information in object tracking remains to be addressed, and thus, few trackers utilize both types of features simultaneously for object matching and localization.

Based on the above analysis, we propose a parallel hybrid architecture for object tracking based on CNN and transformers. CNN has powerful local feature extraction capabilities, while transformers have powerful global feature extraction capabilities. Our model combines local features extracted using CNN with global features extracted using transformers. Object tracking is a problem based on feature matching and correlation, where both global and local features have a certain impact on the performance of the tracking model. Our core idea is to extract local features and global features simultaneously through a parallel dual-branch feature extraction network containing both CNN and transformers. We then achieve the first-stage information interaction between the two branches through our designed bidirectional information interaction channel. Finally, after the fusion of local and global features in the second stage, we perform object tracking and localization. In this work, we not only introduced for the first time a parallel dual-branch feature extraction model by fusing CNN and VIT in the visual object tracking algorithm but also employed depth-wise cross-correlation and a transformer, two information interaction methods, and perfectly fused them.

We evaluate our method on four popular tracking benchmarks, including OTB100 [8], UAV123 [9], LaSOT [10], and TrackingNet [11]. The results show that our tracker achieves state-of-the-art performance on these datasets, with a speed of 43 FPS, which is sufficient for real-time tracking. Our method is able to handle complex scenarios, such as occlusions and changes in appearance, and outperforms traditional CNN-based trackers in terms of both accuracy and speed.

The main contributions of this work can be summarized as follows:We propose an end-to-end object tracking architecture in parallel with the Siamese network and transformer, which only consists of two parts: a dual-branch parallel feature-extraction module and a prediction module. This architecture can effectively combine the local feature extraction capabilities of convolutional neural networks and the global modeling ability of the transformer while keeping the overall architecture simple without complex post-processing, enabling real-time running speed;By combining the features of two different network architectures, the CNN and the transformer, we propose a parallel dual-branch feature extractor that takes advantage of both CNN and transformer feature extraction. It not only obtains local features and global features simultaneously but also allows information exchange between the two branches, exchanging features at different resolutions of local features and global representation, maximally retaining the local features and global representation of the target, significantly improving the target recognizing ability of the tracker;We propose a way of information exchange and feature fusion, achieving two stages of information exchange and feature fusion. The first stage of information exchange is the feature extraction stage, where the local features and global features are exchanged between the two branches, i.e., between the CNN branch and the transformer branch. The second stage of feature fusion is the prediction stage, where the correlation information from the convolutional branch and the search area information from the transformer branch are fused. The fusion of the two stages provides the prediction module with richer feature information;We evaluate our tracker on several large-scale benchmark datasets, including OTB-100 [8], UAV123 [9], LaSOT [10], and TrackingNet [11]. Our tracker achieves state-of-the-art performance on these datasets while also demonstrating its effectiveness in real-time operation.

## 2. Related Work

### 2.1. Visual Object Tracking

Since the introduction of Siamese networks to the field of object tracking, trackers based on Siamese networks have gradually become the mainstream approach in the tracking domain due to their simple architecture and superior performance. SINT [12] was one of the early attempts in offline learning and Siamese networks, pioneering the transformation of object tracking into an image patch-matching problem. SiamFC [6] is considered the starting point of the rise of Siamese network solutions, combining naive correlation with the Siamese framework to create a fast end-to-end tracking network. SiamRPN [13] introduced the region proposal network (RPN) into the Siamese network architecture, transforming the similarity calculation problem into regression and classification problems, thus improving the issue of target scale variation and obtaining more accurate bounding boxes. SiamRPN++ [14] and SiamDW [15] addressed the problem of deep neural networks not being well-suited for tracking tasks using deep networks [16] as feature extractors, further improving tracking performance. SiamMask [17] added an additional mask branch to SiamFC [6], resulting in more accurate detection results and more precise bounding boxes. Algorithms such as SiamBAN [18] and Ocean [19] introduced anchor-free mechanisms [20] from the object detection field into the object tracking domain, improving the accuracy and adaptability of trackers. SiamCAR [21] and SiamFC++ [22] added a center branch in addition to the classification and regression branches to estimate the weights of each bounding box. In addition to optimization work on Siamese architectures, some research has been performed on correlation operations. PG-Net [23] used a pixel-wise cross-correlation method combined with local-to-global correlation matching to suppress background interference. SiamGAT [24] utilized a graph attention mechanism to propagate target information from template features to search features and proposed a target-aware region selection mechanism to adapt to variations in the aspect ratios of different objects.

The main structures of the aforementioned mainstream trackers can be divided into two parts: a backbone network for extracting image features and a correlation-based network for calculating the similarity between the template and search regions. However, these approaches can only capture the local convolutional windows in the template and search regions, neglecting the important global contextual relationships that are crucial for distinguishing the target. As a result, they are prone to being trapped in local optima. Additionally, due to the locality of convolutional operations, these models struggle to capture larger global receptive fields, which may hinder the tracker’s recognition ability. In contrast to the aforementioned approaches, our method combines the advantages of both local and global features while preserving the excellent local feature extraction capability of convolutional neural networks. By incorporating both local and global features, our approach enhances the tracker’s discriminative ability for targets in complex tracking scenarios.

### 2.2. Transformer

The transformer [7], originally proposed by Vaswani et al., was initially used for machine translation tasks and has gradually become the mainstream architecture for language modeling. The transformer model is based on a self-attention mechanism and consists of an encoder and a decoder. It takes a sequence as input, scans each element in the sequence, and learns its dependencies. This characteristic makes the transformer inherently effective at capturing global information in sequential data. Since its introduction, the transformer has achieved tremendous success in the field of natural language processing, such as BERT [25], and has shown remarkable potential in computer vision tasks, including image classification [26], object detection [27], and semantic segmentation [28].

In recent years, researchers have introduced the encoder and decoder architecture of transformers into the field of object tracking, and transformer-based trackers have shown significant effectiveness. Initially, when the transformer was introduced into object tracking, it replaced the correlation modules in Siamese networks and utilized the transformer’s ability to globally integrate features from the template region and search region. For example, TransT [29] introduced a transformer-based feature fusion network composed of multiple self-attention and cross-attention modules. The fused output features were fed into a target classifier and a bounding box regressor. TrTr [30] employed a transformer encoder and decoder as two branches to encode the template features and search features, capturing global information of the target template and search region and using this information to find the correlation between the template and search region. STARK [31], inspired by DETR [27] architecture, treated object tracking as a bounding box prediction problem and designed a transformer-based tracker that modeled the global spatiotemporal feature dependencies between the target object and the search region using self-attention and cross-attention modules. With the development of fully transformer-based models like ViT [26], various object-tracking algorithms completely based on transformers have been proposed. The earliest fully transformer-based tracker, DualTFR [32], segmented the template and search region images into one-dimensional sequences and inputted them into respective transformer feature extraction branches. SwinTrack [33] employed Swin Transformer [34], which has demonstrated excellent performance in object detection as the feature extraction network. It connected the features of the template and search region, along with positional embeddings, and inputted them into the transformer encoder. It utilized cross-attention mechanisms to find the relationship between template and search region features. These algorithms replaced the CNN with the transformer as the feature extraction network while still borrowing from the Siamese architecture. MixFormer [35] was the first single-stage object-tracking algorithm based on the transformer. It employed a set of mixed attention modules (MAM) to simultaneously extract and integrate features from the target template and search region, dramatically improving the performance of the tracker. Similarly, SimTrack [36] and OSTrack [37] also adopted single-stage architectures and further enhanced the tracker’s performance using the MAE [38] pre-training model.

Transformer-based models have been successful in capturing global relationships between the target template and the search region, allowing them to localize strongly correlated regions as the target position in the search area. However, they often overlook the capture of detailed local information from weakly correlated regions. Local relationships also play a significant role in improving object tracking performance and cannot be ignored. While transformer methods establish global correlations, they lack inductive biases. On the other hand, correlation-based methods establish local relationships but lack global modeling and semantic information, making them prone to local optima. By complementing these two types of characteristics, their respective shortcomings can be overcome. Inspired by Conformer [39] and Mobile-former [40], in this work, we propose a parallel feature extraction network that generates more robust and discriminative features for object tracking. This network aims to combine the strengths of both global modeling and local relationships to enhance the tracking performance.

### 2.3. Local-Global Feature Extraction

Local features and global features have equal importance in various downstream tasks in computer vision, and researchers have extensively studied the fusion of these two types of features in the development of computer vision. Conformer [39] proposed a parallel dual-stream network structure consisting of parallel CNN branches and transformer branches. Using feature coupling modules to fuse local and global features, this approach can capture global image information without sacrificing image details. Mobile-former [40] proposed a parallel feature extraction network combining the CNN and transformer, demonstrating excellent performance in downstream tasks such as object detection and instance segmentation. MixFormer [41] proposed a hybrid architecture module combining deep convolution and window-based self-attention in parallel, and this module was used to construct the feature extraction network as the backbone for object detection and segmentation tasks. These studies highlight the significance of integrating local and global features for achieving better performance in computer vision tasks.

The combination of local and global features is also widely applied in the field of object tracking. During the period when correlation filter algorithms were widely used, many tracking algorithms that employed multi-feature fusion emerged. Among them, there were several hybrid trackers that simultaneously utilized local and global features. MUSTer [42] designed a tracker by combining long-term and short-term tracking. The short-term tracking part used an integrated global feature correlation filter (ICF), while the long-term tracking part employed local feature key points matching. Zhao et al. [43] combined the discriminative global features and generative local appearance models to construct a more discriminative and robust tracker. Global features were extracted from the target’s color and texture, and generative local appearance models were obtained using scale-invariant feature transform and spatial geometric information. AMFT [44] integrates multiple types of features, including hand-crafted and deep features, to better model the target appearance in a blur, which is a benefit for tracking in the presence of motion blur. Tao et al. [45] presented a pixel-level supervision neural network (PSNet) to learn discriminative feature representations for forest smoke recognition. This article proposed a model that can extract and fuse local-global features. MS2Net [46] focuses on the task of image super-resolution with heavy motion blur, for which they adopt a network with two branches: one branch for image deblurring and the other one for super-resolution. Its major technical novelties lie in two novel modules: The multi-scale feature fusion module, which fuses features of different scales from the deblurring branch to obtain local and global information, and the multi-stage feature fusion module, which further filters useful information with attention.

During the tracking process, global features and local appearance models were integrated into a Bayesian framework, leading to a more robust and discriminative tracker. The integration of local and global features was mainly applied in tracking algorithms based on the correlation filter framework. However, since the introduction of deep learning into the field of object tracking, there have been few dedicated trackers that simultaneously integrate local and global features specifically for object tracking. Most trackers utilize well-established feature extraction networks that were developed for image classification or detection tasks. Addressing this issue, this study proposes an object-tracking algorithm based on the fusion of CNN and transformer features. A dual-branch parallel feature fusion strategy is designed, where convolutional neural networks (CNN) extract local features and correlation operations establish local relationships, while transformer methods establish global relationships. This approach leverages the complementary advantages of both global and local associations.

## 3. Method

### 3.1. Object Tracking Framework

We propose an end-to-end tracking framework that is composed of only two parts: (1) a feature-extraction module consisting of a fusion of convolutional and transformer features and a multi-head prediction module, where the feature-extraction module is designed in a parallel manner using a CNN feature extraction branch, a transformer feature extraction branch, and bidirectional information exchange channels between the two branches; (2) a dual-head prediction module for target localization.

### 3.2. Feature-Extraction Module

To better utilize local and global features, we designed a dual-branch parallel network structure, as shown in Figure 1. The feature-extraction module consists of a convolutional (CNN) branch, a self-attention (transformer) branch, and bidirectional information exchange channels between the two branches. First, the image is sent to the CNN branch and the transformer branch in the form of feature maps and feature sequences, respectively. Through the bidirectional information exchange channel, local and global features are fully fused to enhance the global perception ability of the CNN branch and enrich the local feature details of the transformer branch, resulting in features with stronger discriminability for the same type of target. Finally, the features output from the two branches are weighted and fused and then sent into the prediction module. Here, the CNN branch adopts the popular Siamese network architecture in recent years, while the transformer branch adopts the one-stage object tracking transformer architecture, which has shown good tracking performance in recent years. This effectively combines the advantages of the two tracking frameworks and is able to obtain more robust features.

Specifically, for the tracking task, we take the first frame of the given video containing the selected target as the template frame, then extract the target template $Z\in R^{H_{z}\times W_{z}\times 3}$ from the first frame, where $H_{z}$ and $W_{z}$ are the height and width of the template image. For the subsequent frames of the given video, they are taken as search frames. The surrounding area of the target template is selected as the search region $X\in R^{H_{x}\times W_{x}\times 3}$, and the target template $Z\in R^{H_{z}\times W_{z}\times 3}$ and the search region $X\in R^{H_{x}\times W_{x}\times 3}$ are input to the feature-extraction module. The two branches of the feature-extraction module process the input image differently. For the transformer branch, before feature extraction, the input image needs to be processed and converted into a feature sequence, similar to the vision transformer. We first crop the target template and the search region into fixed-size image blocks; the image blocks of the target template and the search region are, respectively, denoted as $Z_{p}\in R^{N_{z}\times (P^{2}\cdot 3)}$ and $X_{p}\in R^{N_{x}\times (P^{2}\cdot 3)}$, where P×P is the size of each cropped image, $N_{z}=H_{z}W_{z}/P^{2}$ is the number of images cropped from the target template, and $N_{x}=H_{x}W_{x}/P^{2}$ is the number of images cropped from the search region. Then, each image block is flattened into a one-dimensional feature sequence, followed by the insertion of two groups of position-encoding information markers into the two groups of one-dimensional feature sequences. Finally, we obtain the target template feature sequence $Z_{t}\in R^{N_{z}\times D}$ and the search region feature sequence $X_{t}\in R^{N_{x}\times D}$ with *D* dimensions. After that, we send the target template feature sequence $Z_{t}\in R^{N_{z}\times D}$ and the search region feature sequence $X_{t}\in R^{N_{x}\times D}$ into the transformer branch, where feature extraction and information fusion are performed simultaneously. Afterward, the feature sequences are rearranged and output as a feature image. For the CNN branch, we directly send the original target template $Z\in R^{H_{z}\times W_{z}\times 3}$ and the search region $X\in R^{H_{x}\times W_{x}\times 3}$ to the two sub-branches of CNN. The two sub-branches output φ(Z) and φ(X). Then, we perform a cross-correlation operation on them to obtain the response map R=φ(Z)∗φ(X). Finally, we fuse the response map resulting from the cross-correlation operation in the CNN branch and the feature image rearranged from the transformer branch via weighted summation, and the fused feature image will be sent to the prediction module.

#### 3.2.1. CNN Branch

The design of the CNN branch is based on the Siamese network architecture widely used in the single-target tracking field. The CNN branch includes two sub-branches; one sub-branch takes the target template $Z\in R^{H_{z}\times W_{z}\times 3}$ as input, and the other takes the search region $X\in R^{H_{x}\times W_{x}\times 3}$ as input. These two sub-branches share the same model architecture and parameters, and after feature extraction on the input image, each sub-branch outputs two feature maps φ(Z) and φ(X). Then, φ(Z) and φ(X) are subjected to cross-correlation computation, specifically, the target template $Z\in R^{H_{z}\times W_{z}\times 3}$ is used as a convolution kernel to convolve the search region $X\in R^{H_{x}\times W_{x}\times 3}$ to obtain the correlation response map R between the target template and the search region. In order to retain more position and scale information, we choose deep cross-correlation convolution to extract the correlated features, which was first applied in MobileNet [47] and later widely adopted by other CNN models. The response map R generated by the deep cross-correlation convolution has the same number of channels as φ(Z) and φ(X), which contains a lot of information available for classification and regression. In addition to being able to generate richer correlated features, deep cross-correlation convolution does not produce any hyperparameters and has a much smaller computational cost than traditional convolution operations. Ocean [19], SiamCAR [21], SiamBAN [18], and other methods all use this method to extract correlated features.

Since SiamDW [16] and SiamRPN++ [14] made improvements to the ResNet-50 [48] network for target tracking, ResNet-50 has gradually become the backbone network of most target-tracking algorithms. Even transformer-based target-tracking algorithms such as TrDiMP [49], TransT [29], and STARK [31] use ResNet-50 [48] as the feature extraction network. We borrowed the main structure of ResNet-50 [48] but made some lightweight modifications to its internal structure. As shown in Table 1, we only borrowed the first four stages of the standard ResNet-50 [48], removed the final stage’s convolution and fully connected layers, and adjusted the downsampling unit’s stride in the fourth stage from 2 to 1. Then, we referenced the literature ResNeXt [50], using grouped convolution to improve performance by dividing the convolution channels into groups to reduce computational cost. The saved computational resources were used to increase the network’s feature channel. To fully utilize the saved computational power, we increased the base channel of ResNet-50 [48] from the original 64 to 128, which is the same as ResNeXt-50 [50]. Next, we used the inverted residual bottleneck module, which was first introduced in the literature MobileNetV2 [51], followed by a series of lightweight neural networks based on the neural architecture search technique NAS [52] in the literature EfficientNet [53] and the latest state-of-the-art model ConvNeXt [54], which all adopted the inverted bottleneck structure. The main reason for choosing the inverted bottleneck module is twofold: first, it improves feature extraction performance, and second, it reduces the output channel of each residual block, which reduces the computational cost of information interaction channels between the transformer branch.

#### 3.2.2. Transformer Branch

The design of the transformer branch partially references the widely applied ViT [26] algorithm’s network architecture, which consists of multiple transformer encoders, each composed of a multi-head self-attention (MSA) module and a multi-layer perceptron (MLP) module. Layernorm operations are performed before each multi-head self-attention (MSA) module and multi-layer perceptron (MLP) module, and each multi-head self-attention (MSA) module and multi-layer perceptron (MLP) module also includes a residual connection. Another important advantage of borrowing from the ViT algorithm is that researchers have provided a large number of pre-trained models based on the ViT architecture. The use of these publicly available pre-trained models can greatly reduce our training costs.

The input of the transformer branch, as mentioned in the previous text, first processes the input target template $Z\in R^{H_{z}\times W_{z}\times 3}$ and the search region $X\in R^{H_{x}\times W_{x}\times 3}$ via serialization, converting them into a one-dimensional target template sequence $Z_{t}\in R^{N_{z}\times D}$ and search region sequence $X_{t}\in R^{N_{x}\times D}$. We connect them along the one-dimensional direction and send them into the transformer encoder. The forward propagation process of the transformer branch can be represented as:(1)$$\left[\begin{array}{c} Z_{t}^{*}\\ X_{t}^{*} \end{array}\right]=\left[\begin{array}{c} Z_{t}^{n}\\ X_{t}^{n} \end{array}\right]+ATT\left[\begin{array}{c} Z_{t}^{n}\\ X_{t}^{n} \end{array}\right],\;\left[\begin{array}{c} Z_{t}^{n+1}\\ X_{t}^{n+1} \end{array}\right]=\left[\begin{array}{c} Z_{t}^{*}\\ X_{t}^{*} \end{array}\right]+MLP\left[\begin{array}{c} Z_{t}^{*}\\ X_{t}^{*} \end{array}\right],$$
where $Z_{t}^{n}$ and $Z_{t}^{n+1}$ represent the target template sequence of the ${\left(n\right)}_{th}$ layer and the ${\left(n+1\right)}_{th}$ layer, $X_{t}^{n}$ and $X_{t}^{n+1}$, respectively, represent the search region sequence of the ${\left(n\right)}_{th}$ layer and the ${\left(n+1\right)}_{th}$ layer. This design allows the target template sequence $Z_{t}\in R^{N_{z}\times D}$ and the search region sequence $X_{t}\in R^{N_{x}\times D}$ to interact early on, unlike the CNN branch, which simultaneously achieves feature extraction and feature interaction within one transformer branch, so there is no need to add an additional feature interaction module after the backbone network of the transformer branch. Recently, single-stage target-tracking algorithms with good tracking results, such as MixFormer [41], OSTrack [37], SimTrack [36], and ProContEXT [55], all use this design. However, unlike the aforementioned algorithms, the transformer branch in this study rearranges and combines the feature sequences $Z_{t}$ and $X_{t}$ after each encoder output (reshape) into the form of feature maps and, through an information interaction module, the target template image block interacts with the target template feature map $\varphi^{*}(Z)$ from the CNN branch, and the search region image block interacts with the search region feature map $\varphi^{*}(X)$ from the CNN branch, as shown in Figure 2. After the feature extraction is completed, the feature sequence of the search region sequence $X_{t}$ output by the transformer branch is rearranged into the form of a feature map and then sent together with the output of the CNN branch’s cross-correlation feature R for weighted fusion. After fusion, they are sent to the prediction module together.

### 3.3. Bidirectional Information Interaction Module

The multi-channel feature maps output from the CNN branch and the one-dimensional feature sequences output from the transformer branch are two different forms of feature representation. Effectively merging the features with different representation forms is an important problem. To solve this problem, we designed a bidirectional information exchange module, allowing the local features from the CNN branch and the global features from the transformer branch to interact with each other during the feature extraction process.

#### 3.3.1. CNN→Transformer Channel

We must be aware that the feature dimensions of CNN and transformer are inconsistent. The representation dimension of a CNN feature map is generally H × W × C, where H, W, and C represent the height, width, and number of channels of the feature map, respectively. On the other hand, the dimension of a transformer feature sequence is K × D, where K and D represent the number of feature sequences and the encoding dimension, respectively. In order to facilitate information interaction between the two branches of CNN and the transformer, we designed a bidirectional information interaction module.

Compared to the transformer, the CNN branch’s features contain richer spatial position information. Therefore, for the bidirectional information interaction channel between the CNN branch and the transformer branch, we used the spatial attention mechanism from CBAM [56]. Specifically, the template sequence $Z_{t}\in R^{N_{z}\times D}$ and the search area sequence $X_{t}\in R^{N_{x}\times D}$ from the transformer branch retain their original feature sequence numbers $\left(N_{z},N_{x}\right)$ and encoding dimensions D, and are rearranged into the template feature map $Z_{p}^{*}\in R^{H_{z}^{*}\times W_{z}^{*}\times D}$ and the search area feature map $X_{p}^{*}\in R^{H_{x}^{*}\times W_{x}^{*}\times D}$, where $H_{z}^{*}=W_{z}^{*}=\sqrt{N_{z}}$, $H_{x}^{*}=W_{x}^{*}=\sqrt{N_{x}}$. To align the sizes of the CNN branch’s feature map with the transformer branch’s feature map, the target template feature map $Z_{C}\in R^{{\tilde{H}}_{z}\times {\tilde{W}}_{z}\times \tilde{C}}$ and the search area feature map $X_{C}\in R^{{\tilde{H}}_{x}\times {\tilde{W}}_{x}\times \tilde{C}}$ from the CNN branch are downsampled once by pooling to match the size of the feature map from the transformer branch. Then, the channel dimension of the CNN feature map is processed with max pooling and average pooling, respectively, to output $Z_{s}^{*}\in R^{H_{z}^{*}\times W_{z}^{*}\times 2}$ and $X_{s}^{*}\in R^{H_{x}^{*}\times W_{x}^{*}\times 2}$, followed by a convolutional operation to output $Z_{s}^{}\in R^{H_{z}^{*}\times W_{z}^{*}\times 1}$ and $X_{s}^{}\in R^{H_{x}^{*}\times W_{x}^{*}\times 1}$. Finally, the transformed template feature map $Z_{p}^{*}\in R^{H_{z}^{*}\times W_{z}^{*}\times D}$ and the search area feature map $X_{p}^{*}\in R^{H_{x}^{*}\times W_{x}^{*}\times D}$ are processed in the same way as the spatial attention mechanism in CBAM [56]. In this way, the transformer branch can successfully incorporate spatial features from the CNN branch. The combined features are then rearranged into template sequence $Z_{t}\in R^{N_{z}\times D}$ and search area sequence $X_{t}\in R^{N_{x}\times D}$ and sent to the next encoder of the next transformer branch, as shown in Figure 3. Although our design is similar to the spatial attention in CBAM [56], they still have significant differences. The most important one is that our channel interaction takes input from the parallel branches of the transformer and CNN, while the spatial attention mechanism in CBAM [56] only performs calculations within a single CNN branch. In our design, the CNN→Transformer channel uses a twin structure similar to that of the CNN branch, where the CNN→Transformer channel for interacting with the template and the CNN→Transformer channel for interacting with the search area share the same structure and parameters.

#### 3.3.2. Transformer→CNN Channel

Compared to CNN features, the output of the transformer branch has a larger feature sequence dimension, and each one-dimensional feature sequence contains richer information. To incorporate the more abundant global information from the transformer branch into the local features of the CNN branch, we drew inspiration from SENet [57] to design a Transformer→CNN channel. As shown in Figure 4, first, like the CNN→Transformer channel, the target template sequence $Z_{t}\in R^{N_{z}\times D}$ and the search area sequence $X_{t}\in R^{N_{x}\times D}$ from the transformer branch are rearranged into the target template feature map $Z_{p}^{*}\in R^{H_{z}^{*}\times W_{z}^{*}\times D}$ and the search area feature map $X_{p}^{*}\in R^{H_{x}^{*}\times W_{x}^{*}\times D}$. Next, they go through a global average pooling (GAP) to obtain two 1D features containing global information, $Z_{s}^{*}\in R^{1\times 1\times D}$ and $X_{s}^{*}\in R^{1\times 1\times D}$. Then, these features pass through two consecutive 1 × 1 convolutional layers, with a BN layer used for regularization between the convolutional layers. Finally, after using a GELU activation function, the output 1D features $Z_{s}^{}\in R^{1\times 1\times \tilde{C}}$ and $X_{s}^{}\in R^{1\times 1\times \tilde{C}}$ are the same as the channel number of the CNN branch, and they are multiplied channel-wise with the target template feature map $Z_{C}\in R^{{\tilde{H}}_{z}\times {\tilde{W}}_{z}\times \tilde{C}}$ and the search area feature map $X_{C}\in R^{{\tilde{H}}_{x}\times {\tilde{W}}_{x}\times \tilde{C}}$ from the CNN branch’s output. After the multiplication, the features are added to the original feature map of the CNN branch and are then sent into the next convolutional module of the CNN branch. It is worth mentioning that our designed Transformer→CNN channel changes the dimension of the feature channel, which is different from SENet [57]. Figure 5 shows a complete block diagram of the target template or search area going through the feature-extraction module.

### 3.4. Feature Fusion

As known from Section 3.1, the CNN branch not only needs to extract features but also perform cross-correlation operations between the template and the search region. To reduce the computational cost, we crop the edges of the 16 × 16 target template, resize it to 8 × 8, and then perform cross-correlation with the 32 × 32 search region. Similarly to SiamRPN++ [14], SiamBAN [18], and SiamCAR [21], the cross-correlation operation is performed using depth-wise convolution.

It should be clarified that the response map output from the CNN branch after cross-correlation and the feature map output from the transformer branch have different sizes; therefore, they cannot be directly added, concatenated, or fused with weighting. To ensure the consistency of the size of the feature map after cross-correlation with the size of the search region feature map, enabling feature fusion, we need to pad the search region before performing cross-correlation. Since the size of the target is even, using an even convolution kernel for the operation cannot achieve symmetric padding of the search region. Therefore, to ensure the accuracy of the subsequent prediction, we adopt the method from [58], dividing the feature map of the search region into four groups and padding one row/column in each of the top-left, bottom-left, top-right, and bottom-right directions, as shown in Figure 6. Then, we perform deep cross-correlation between the padded search region and the cropped target template. At this point, the response map after cross-correlation is four times the size of the Transformer branch’s output search region feature map, and after downsampling through a single pooling operation, it can be weighted and fused with the search region output from the transformer branch. The response map output from the CNN branch after cross-correlation and the feature map output from the transformer branch are then weighted and concatenated before being sent to the prediction module, as shown in Figure 7.

### 3.5. Prediction Module

#### 3.5.1. Double-Head Predictor

Existing trackers mostly use fully connected networks or convolutional networks to classify foreground and background, regress the target bounding box, and do not deeply analyze or design networks based on the characteristics of classification and regression tasks. Inspired by [59], which reinterprets the classification and localization subtasks in the detection task, it is found that the fc-head is more suitable for the classification task, while the conv-head is more suitable for the localization task. Therefore, we designed a dual-head predictor to improve the accuracy of classification and regression. Specifically, the classification branch of the prediction module consists of two fully connected layers, while the regression branch consists of three convolutional layers. To make the tracking framework more concise, the method in this study directly predicts normalized coordinates, abandoning the use of anchor points and anchor boxes based on prior knowledge. The fused features are directly input into the prediction heads to predict the classification results of foreground/background and the normalized coordinates of the search region size.

#### 3.5.2. Training Loss

For the classification task, we use the cross-entropy loss to calculate the classification loss (denoted as $L_{{}_{cls}}$). For the regression task, we use a linear combination of L1 loss and GIoU loss [60] (denoted as $L_{GIoU}$). The total loss function is denoted as:(2)$$L=\lambda_{cls}L_{cls}+\lambda_{1}L_{1}+\lambda_{GIoU}L_{GIoU}$$
where $\lambda_{cls}$, $\lambda_{1}$, and $\lambda_{GIoU}$ represent the regularization parameters for classification loss, L1 loss, and GIoU loss. In practice, these parameters are set to $\lambda_{cls}=1$, $\lambda_{1}=2$, and $\lambda_{GIoU}=5$.

## 4. Experiments

### 4.1. Training

The training process of the proposed model consists of two stages. In the first stage, we pre-train the feature-extraction module on the large-scale ImageNet-1K [61] dataset for classification tasks. This dataset contains 1.28 million training images from 1000 categories. We use the ViT-Base pre-trained by MAE as the initial parameters for the transformer branch of the feature-extraction module, while the remaining parameters are initialized using the Xavier [62] Init method. For the network optimizer, we choose the AdamW [63] optimizer, with the pre-trained weight learning rate set to $10^{-5}$, and the rest set to $10^{-4}$. The feature-extraction module is trained for a total of 100 epochs.

After completing the pre-training of the feature-extraction module, we train the entire target tracking model on the tracking dataset using COCO [64], TrackingNet [11], LaSOT [10], and GOT-10k [65]. For the video datasets TrackingNet [11], LaSOT [10], and GOT-10k [65], we directly sample images from the video sequences to collect the training samples. For the COCO dataset [64], we use some common data augmentation methods like translation to enlarge the training set. We load the parameters of the pre-trained feature-extraction module and initialize the remaining parameters using the Xavier Init method [62]. We choose AdamW [63] as the network optimizer, with the learning rate for the feature-extraction module set to $10^{-5}$, the learning rate for other parameters set to $10^{-4}$, and the weight decay rate set to $10^{-4}$. The proposed method was implemented in Python3.8 using the PyTorch1.11.0 framework on a server with four NVIDIA TITAN X GPUs. The model was trained for a total of 300 epochs, with each epoch having 1000 iterations, and the batch size was set to 256. The learning rate was reduced to 1/10 of its original value after 200 epochs.

### 4.2. Evaluation

To further evaluate the practical effectiveness of the proposed target tracking model, we conducted experiments on the testing sets of OTB100 [8], UAV123 [9], TrackingNet [11], and LaSOT [10]. We also compared the results with some of the representative tracking algorithms such as DropTrack [61], OSTrack [37], MixFormer [41], STARK [31], TransT [29], SiamGAT [24], SiamAttn [62], SiamR-CNN [63], and SiamBAN [18]. We provided detailed results of success rate, precision, and normalized precision, where success rate indicates the percentage of frames with the overlap between the predicted and ground-truth bounding boxes above a given threshold, precision represents the percentage of frames with the distance between the predicted and ground-truth bounding boxes below a given threshold, and normalized precision takes the true bounding box into account in calculating the precision, while only considering success rate and precision on the OTB2015 [8] and UAV123 [9] datasets.

#### 4.2.1. Results on OTB100 Benchmark

The OTB100 [8] dataset contains 100 video sequences and is one of the earlier single-object tracking benchmarks. Each video sequence in the dataset is annotated with attributes, including fast motion, low resolution, and illumination variation, which are challenging attributes. Figure 8 shows a comparison of the success rate and precision of our algorithm and several other algorithms on the OTB100 dataset. Our tracker achieves a performance of 92.6% in precision and 71.5% in success rate on the OTB100 dataset.

Since most OTB videos have fewer frames and the appearance features of the targets remain relatively unchanged in many sequences, trackers based on CNN feature extraction and correlation matching show good tracking performance. However, the performance of one-stage transformer-based trackers is highly dependent on their attention mechanism and ability to capture global features, thus scoring low. As shown in Table 2, our proposed parallel tracking algorithm combines the local feature extraction and correlation matching capabilities of CNN with the global feature capture ability and attention mechanism of the transformer, achieving the best results on the OTB100 test set. Figure 9 shows the visualized results of our tracker on some video sequences in the OTB100 test set.

#### 4.2.2. Results on UAV123 Benchmark

UAV123 [9] is a benchmark dataset containing 123 fully annotated high-definition videos captured from a low-altitude aerial perspective, including 103 stable video sequences, 12 unstable video sequences, and 8 composite video sequences. All sequences are high-definition videos captured via low-altitude drones and fully annotated. Figure 10 shows a comparison of the success rate and precision of our tracker on the UAV123 dataset.

Tracking targets on the UAV123 dataset is more challenging than on other benchmark datasets because the target objects are relatively small in aerial tracking sequences, resulting in limited visual information that can be obtained. Additionally, the target objects and the camera often change their positions and orientations. As can be seen from Table 3, our tracker achieves the best tracking results on the UAV123 dataset.

#### 4.2.3. Results on LaSOT Benchmark

LaSOT [10] is a large-scale, high-quality target-tracking dataset, with each frame carefully reviewed and manually annotated. The dataset contains 1400 video sequences, including 1120 training sequences and 280 test sequences, with an average sequence length of over 2500 frames, divided into 70 target categories, each consisting of 20 video sequences, encompassing various challenges. The tracking results on the LaSOT dataset are shown in Table 4, where “—” indicates that the experimental results are not provided in the original paper.

#### 4.2.4. Results on TrackingNet Benchmark

TrackingNet [11] is a large-scale target-tracking dataset covering various scenarios in natural scenes, including various frame rates, resolutions, contextual scenes, and target categories. The training set of this dataset is divided into 12 training subsets, each containing 2511 video sequences, with each video sequence having approximately 400 frames. It is meant to be used for training and testing short-term tracking algorithms. Its validation set contains 511 video sequences. The tracking results on the TrackingNet dataset are shown in Table 4.

### 4.3. Ablation Study

To validate the effectiveness of the modules in our proposed method, ablation experiments were conducted on the OTB100 [8], UAV123 [9], and LaSOT [10] datasets. The success rate and precision (normalized precision for LaSOT) were used to evaluate the experimental results.

To validate the effectiveness of different components in the parallel CNN-transformer feature-extraction module, we designed different variants of CVTrack. Table 5 shows a comparison of the results for these different variants. CVTrack refers to the complete algorithm proposed in this study, CVTrack-v1 refers to the algorithm without the transformer branch in the feature-extraction module, CVTrack-v2 refers to the algorithm without the CNN branch in the feature-extraction module, and CVTrack-v3 refers to the algorithm without the feature interaction channel in the feature-extraction module.

As shown in the table, the modified three algorithms are compared with the CVTrack algorithm in terms of success rate and precision (normalized precision) on the three datasets, and then the average is taken. The CVTrack algorithm is 5.8% and 5.4% higher than the CVTrack-v1 algorithm, 2.6% and 2.1% higher than the CVTrack-v2 algorithm, and 1.0% and 0.9% higher than the CVTrack-v3 algorithm. This indicates that the different components in the feature-extraction module will affect the final tracking performance. The feature-extraction module with a parallel dual-branch architecture performs better than a single-branch CNN feature-extraction module or a single-branch transformer feature-extraction module. Secondly, the feature-extraction module with an information interaction channel performs better than a simple dual-branch parallel feature-extraction module. Finally, compared to the CNN branch, the transformer branch shows better performance, indicating that the feature-extraction effect of the transformer branch is better than that of the simple CNN branch.

### 4.4. Qualitative Comparison

Figure 9 shows the results of the visual comparison between our method and other state-of-the-art tracking methods on some test video sequences. The SiamGAT [24] tracking method based on CNN performs easily disturbed by the fast motion and scale variation attributes. The Stark [31] tracking algorithm uses CNN for feature extraction and then uses a transformer for similarity calculation. The Mixformer [41] tracking algorithm using the one-stage transformer method obtains some accurate tracking results on the OTB100 test video sequences due to the transformer model that has been usefully adopted. However, the tracking results of the tracking method based on the transformer on other test video sequences disturbed by similar objects are still unacceptable. Compared to other tracking methods, the proposed CVTrack method could accurately track these targets in complex tracking scenarios, which verified that the proposed featured fusion model is fully effective.

### 4.5. Failure Cases

Figure 11 shows some failure cases of the proposed CVTrack. To display the tracking results more intuitively, we also give the ground-truth label of the target as a reference. For the Ironman testing sequence, the main reason why the proposed tracker cannot track the target is the challenge of low illumination and low resolution. For the soccer testing sequence, due to the influence of illumination variation and occlusion, our tracker lost the target. For these failure tracking cases, we will further explore them in future work.

## 5. Conclusions

This study proposes an end-to-end parallel tracking model that integrates CNN’s local feature extraction ability and the transformer’s global feature extraction ability. We use convolution and self-attention mechanisms to collaboratively process feature images while integrating local and global information about feature images. Specifically, the convolutional neural network branch is mainly used to extract local information from images, while the transformer branch’s self-attention mechanism is mainly used to calculate the global information of images. In addition, we design a bidirectional information interaction module to fuse local features from the CNN branch and global features from the transformer branch through information interaction. Finally, through the dual-head prediction module, our tracker not only learns task-related features but also improves performance. Extensive experiments show that our tracker achieves state-of-the-art performance and real-time running speed. Our work demonstrates that the parallel interaction design of the transformer and CNN provides a new direction for the tracking field.

## Figures and Tables

**Figure 1 sensors-24-00274-f001:**
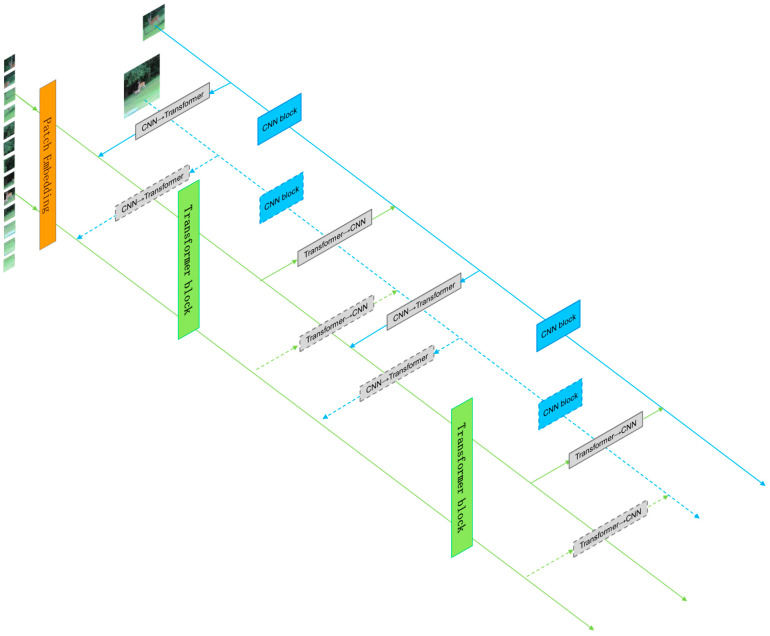
Tracking algorithm architecture.

**Figure 2 sensors-24-00274-f002:**
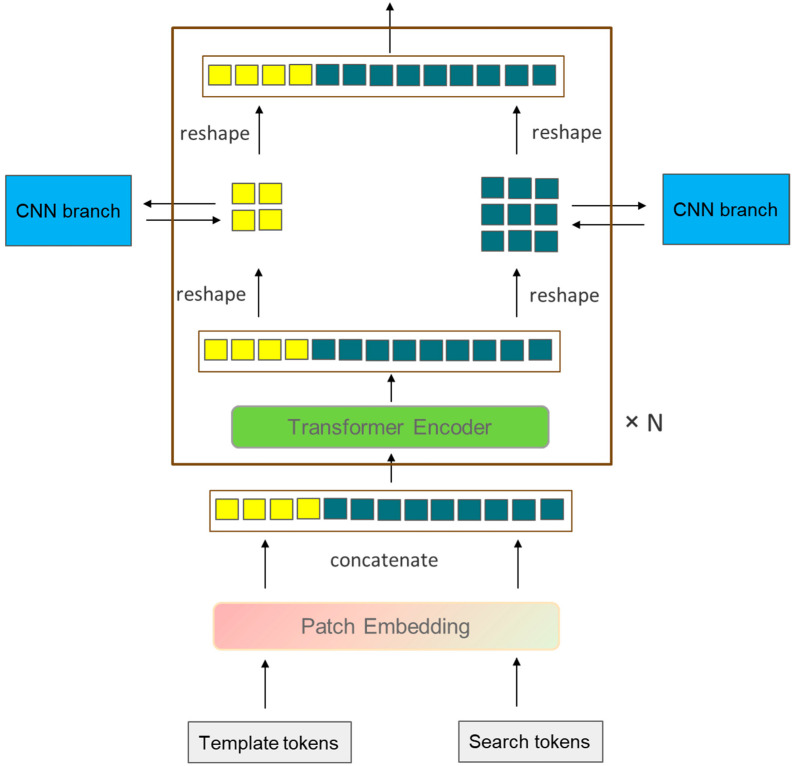
A simplified block within the transformer branch (with the CNN branch’s structure simplified for clarity).

**Figure 3 sensors-24-00274-f003:**
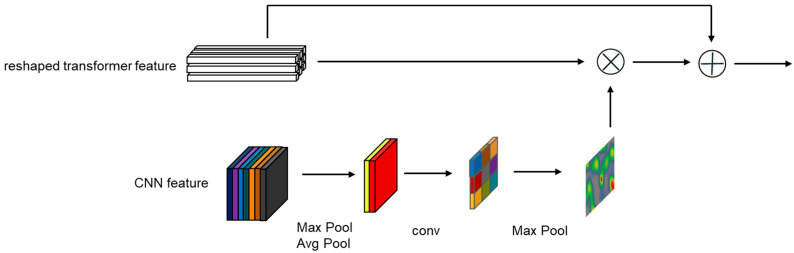
Architecture of a CNN→Transformer channel unit.

**Figure 4 sensors-24-00274-f004:**
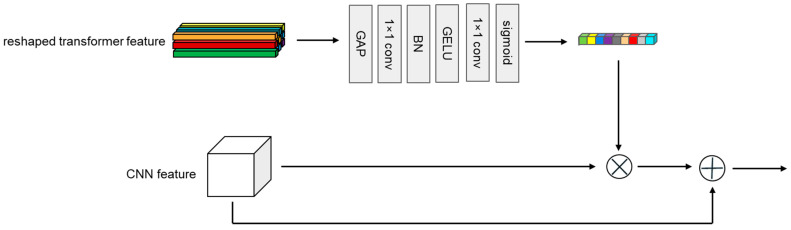
Architecture of a Transformer→CNN channel unit.

**Figure 5 sensors-24-00274-f005:**
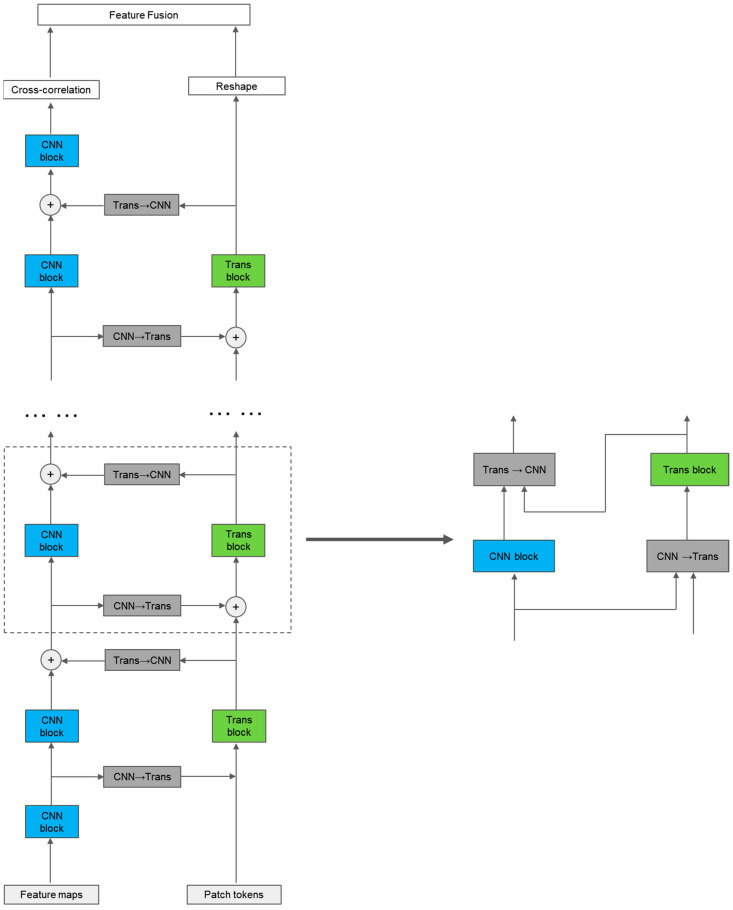
A complete module forward propagation diagram of the feature-extraction module within the target template or search area, including a CNN block, a transformer block, a CNN→Transformer block, and a Transformer→CNN block.

**Figure 6 sensors-24-00274-f006:**
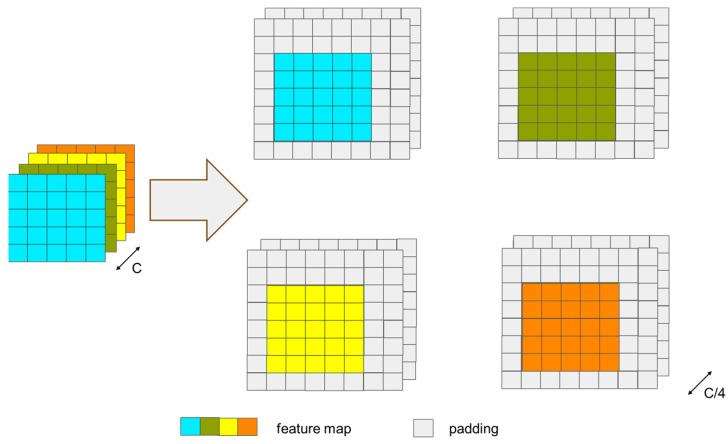
The figure of feature map padding for the boundary.

**Figure 7 sensors-24-00274-f007:**
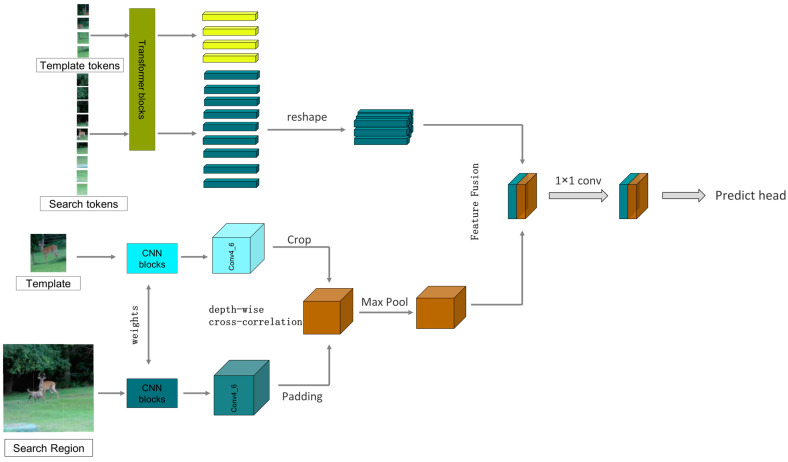
The figure of feature fusion (without showing the feature interaction channels).

**Figure 8 sensors-24-00274-f008:**
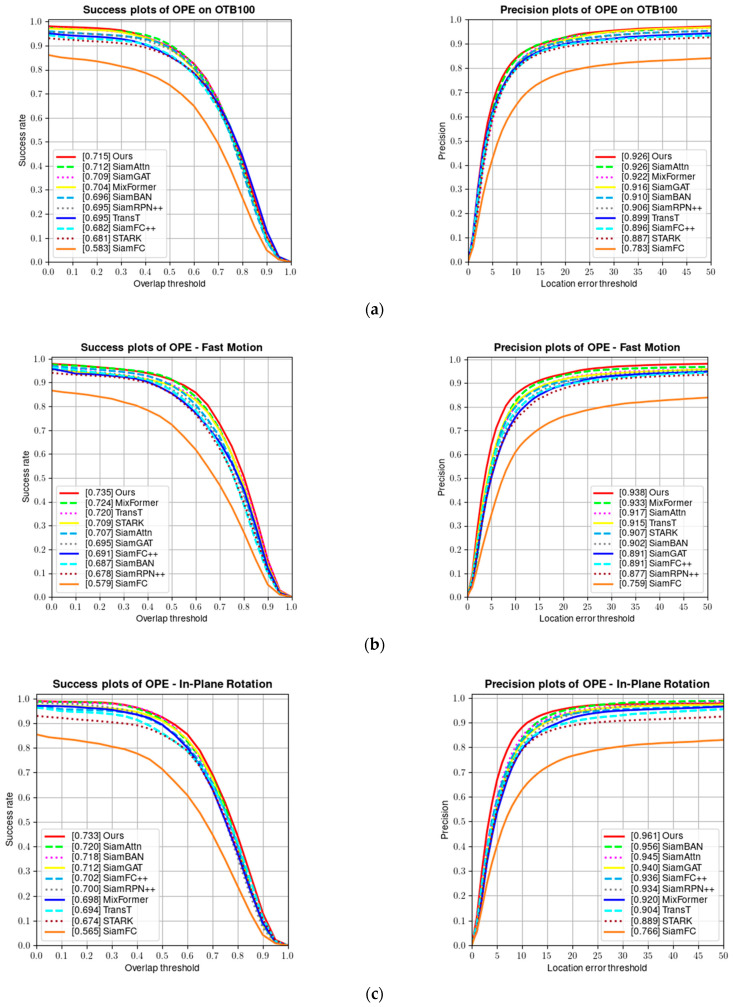
(**a**) Success and precision plots on the OTB100 dataset; (**b**) success and precision rate under fast motion; (**c**) success and precision rate under in-plane rotation; (**d**) success and precision rate under low resolution; and (**e**) success and precision rate under illumination variation.

**Figure 9 sensors-24-00274-f009:**
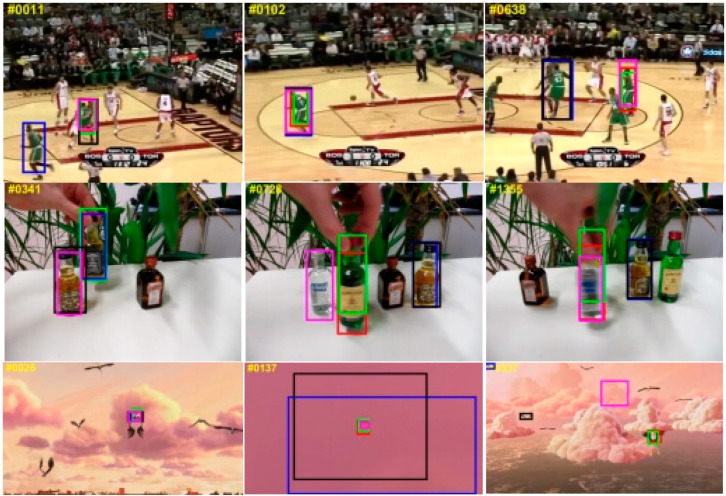
Visualization of tracking results for some sequences in the OTB100 dataset. Red boxes are our model results, green boxes are ground truth, blue boxes are results from Mixformer, black boxes are results from STARK, and pink boxes are results from SiamGAT.

**Figure 10 sensors-24-00274-f010:**
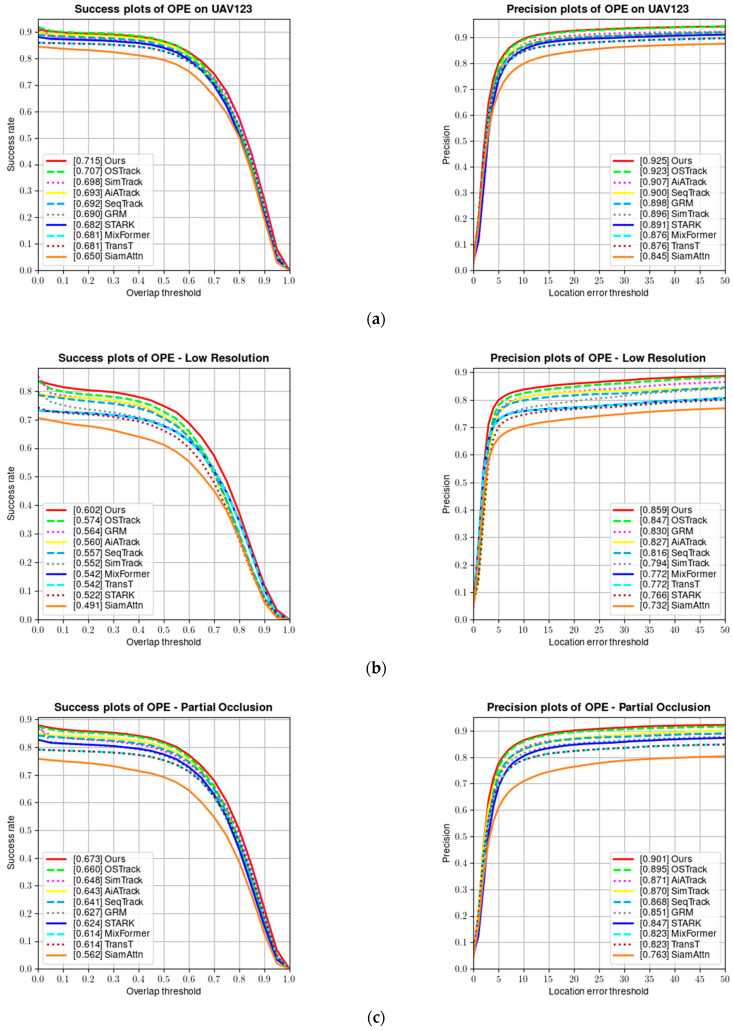
(**a**) Success and precision plots on the UAV123 dataset; (**b**) success and precision rate under low resolution; (**c**) success and precision rate under partial occlusion; (**d**) success and precision rate under viewpoint change; (**e**) success and precision rate under scale variation; and (**f**) success and precision rate under full occlusion.

**Figure 11 sensors-24-00274-f011:**
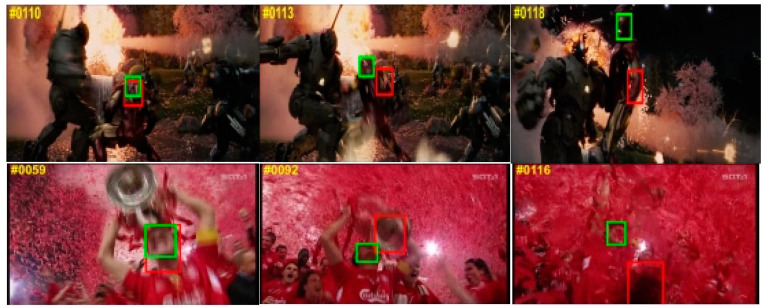
Failure cases (from top to bottom are Ironman and soccer). The proposed CVTrack results are shown in red boxes, and the target ground truth is shown in green boxes.

**Table 1 sensors-24-00274-t001:** Architecture of feature-extraction module.

	CNN Branch	InformationExchange Channel	Transformer Branch	
stage	Output size	structure		structure	Output size	
c1	S: 128 × 128T: 64 × 64	7 × 7, 64Stride = 2	Patch Embedding	S: 256 × 768T: 64 × 768	×1
S: 64 × 64T: 32 × 32	3 × 3 max pooling, Stride = 2
c2	S: 64 × 64T: 32 × 32	1 × 1, 643 × 3, 256, C = 321 × 1, 64	CNN→Trans	MHSA-12, 7681 × 1, 3072 (MLP hidden size)1 × 1, 768	S: 256 × 768T: 64 × 768	×3

CNN←Trans
c3	S: 32 × 32T: 16 × 16	1 × 1, 1283 × 3, 512, C = 321 × 1, 128	CNN→Trans	MHSA-12, 7681 × 1, 3072 (MLP hidden size)1 × 1, 768	S: 256 × 768T: 64 × 768	×4

CNN←Trans
c4	S: 32 × 32T: 16 × 16	1 × 1, 2563 × 3, 1024, C = 321 × 1, 256	CNN→Trans	MHSA-12, 7681 × 1, 3072 (MLP hidden size)1 × 1, 768	S: 256 × 768T: 64 × 768	×5

CNN→Trans
S: 32 × 32T: 16 × 16	1 × 1, 2563 × 3, 1024, C = 321 × 1, 256				×1
	32 × 32	depth-wise cross-correlated	Rearrange	S: 16 × 16T: 8 × 8	×1
fusion	16 × 16	2 × 2 max pooling, Stride = 2	Feature fusion	Extract search region	16 × 16	×1

**Table 2 sensors-24-00274-t002:** Comparison on OTB100 [8] dataset. Top three results are shown in red, green, and blue fonts, respectively.

Methods	Source	Backbone	AUC	Prec
SiamFC	ECCVW2016	CNN	58.3	78.3
SiamRPN++	CVPR2019	CNN	69.6	91.5
SiamBAN	CVPR2020	CNN	69.6	91.0
SiamR-CNN	CVPR2020	CNN	70.1	89.1
SiamAttn	CVPR2020	CNN	71.2	92.6
SiamGAT	CVPR2021	CNN	71.0	91.6
TransT	CVPR2021	CNN	69.5	89.9
STARK	ICCV2021	CNN	68.0	88.4
SimTrack-B/16	ECCV2022	Transformer	66.1	85.7
Mixformer-L	CVPR2022	Transformer	70.4	92.2
Ostrack-384	ECCV2022	Transformer	68.1	88.7
DropTrack	CVPR2023	Transformer	69.6	90.0
CVTrack	Ours	CNN + Transformer	71.5	92.6

**Table 3 sensors-24-00274-t003:** Comparison of UAV123 [9] dataset. Top three results are shown in red, green, and blue fonts, respectively.

Methods	Template Size	Search Size	AUC	Prec
SiamFC	127 × 127	255 × 255	48.5	69.3
SiamRPN++	127 × 127	255 × 255	64.2	84.0
SiamFC++_GoogLeNet	127 × 127	303 × 303	62.3	81.0
SiamBAN	127 × 127	255 × 255	63.1	83.3
SiamAttn	127 × 127	255 × 255	65.0	84.5
TransT	128 × 128	256 × 256	68.1	87.6
STARK	128 × 128	320 × 320	68.5	89.5
SimTrack-B/16	112 × 112	224 × 224	69.8	89.6
SimTrack-B/14	112 × 112	224 × 224	71.2	91.6
Mixformer	128 × 128	320 × 320	68.7	89.5
Mixformer-L	128 × 128	320 × 320	69.5	91.0
Ostrack-256	128 × 128	256 × 256	68.3	88.8
Ostrack-384	192 × 192	384 × 384	70.7	92.3
DropTrack	192 × 192	384 × 384	70.9	92.4
Ours	128 × 128	256 × 256	71.4	92.5

**Table 4 sensors-24-00274-t004:** Comparison of LaSOT [10] and TrackingNet [11] datasets. Top three results are shown in red, green, and blue fonts, respectively.

Methods	LaSOT			TrackingNet		
	AUC	Norm.Prec	Prec	AUC	Norm.Prec	Prec
SiamRPN++	49.6	56.9	49.1	73.3	80.0	60.4
SiamBAN	51.4	59.8	52.1	—	—	—
SiamAttn	56.0	64.8	—	75.2	81.7	—
TransT	64.9	73.8	69.0	81.4	86.7	80.3
STARK	67.1	76.9	72.2	82.0	86.9	79.1
SimTrack	69.3	78.5	74.0	82.3	86.5	80.2
Mixformer	70.0	79.9	76.3	83.9	88.9	83.1
Ostrack	71.1	81.1	77.6	83.9	88.5	83.2
Ours	70.7	80.1	76.1	83.4	88.0	83.4

**Table 5 sensors-24-00274-t005:** Comparison of the results from ablation experiments conducted on multiple datasets. AUC-O and Prec-O represent the success rate and precision on the OTB100 benchmark, AUC-U, and Prec-U represent the success rate and precision on the UAV123 benchmark, and AUC-L and N.Prec-U represent the success rate and normalized precision on the LaSOT benchmark. FPS represents the testing speed on the UAV123 benchmark.

Methods	AUC-O	Prec-O	AUC-U	Prec-U	AUC-L	N.Prec-L	FPS
CVTrack	71.5	92.6	71.5	91.9	70.7	80.1	43
CVTrack-v1	69.1	90.9	63.4	84.9	63.7	72.5	71
CVTrack-v2	68.3	89.8	68.6	90.0	69.0	78.4	58
CVTrack-v3	70.3	91.6	70.2	91.1	70.1	79.0	47

## Data Availability

Data is contained within the article.

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
