# Peer review of "CVTrack: Combined Convolutional Neural Network and Vision Transformer Fusion Model for Visual Tracking"

_sensors, 2024, doi:10.3390/s24010274_

Round 1

Reviewer 1 Report

Comments and Suggestions for Authors

This paper proposes a CVTrack that leverages a parallel dual-branch backbone network combining CNN and Transformer for feature extraction and fusion. Overall, the structure of this paper is clear. The main pipeline of this paper is easy to follow. However, there are still issues needed to be clarified:

1. The problem definition and motivation must be clearly stated. The introduction section should be more comprehensive. Also, please show the main differences between the proposed work and the recently published related works.

2. Experiment validations are not convincing. Some additional experiments need to be conducted to make its conclusion stronger.

3. Some most related works should be discussed in this paper:  Robust thermal infrared tracking via an adaptively multi-feature fusion model, NCAA, 2023

4. Please consider discussing and analyzing the limitations of the proposed method.

5. Some of your figures need improvement, such as figures 1,3,5.

Comments on the Quality of English Language

The language is more or less correct, but the level could be improved. There are some typos, and grammar errors that need to be corrected.

Author Response

1.Summary

Thank you very much for taking the time to review this manuscript. Please find the detailed responses below and the corresponding corrections highlighted in the re-submitted files.

2.Point-by-point response to Comments and Suggestions for Authors

Comments 1: The problem definition and motivation must be clearly stated. The introduction section should be more comprehensive. Also, please show the main differences between the proposed work and the recently published related works.

Response 1: Firstly, we have added the explain of definition and motivation in line 73-82. Then we have enriched the introduction section in line 88-91. Also, we have shown the main differences between the proposed work and the recently published related works in line 106-109. The changes have been highlighted in the re-submitted files.

Comments 2: Experiment validations are not convincing. Some additional experiments need to be conducted to make its conclusion stronger.

Response 2: We have read your recommended reference Robust thermal infrared tracking via an adaptively multi-feature fusion model and compared it with it. Then, we have added two parts Qualitative comparison and Failure cases in Experiments section. These changes have been highlighted in line 690-719.

Comments 3: Some most related works should be discussed in this paper:  Robust thermal infrared tracking via an adaptively multi-feature fusion model, NCAA, 2023.

Response 3: We have read this paper and discussed it in our article in line 259-261.

Comments 4: Please consider discussing and analyzing the limitations of the proposed method. 

Response 4: We have discussed and analyzed the limitations of the proposed method. Then we have added the Failure cases in Experiments section in line 709-719.

Comments 5: Some of your figures need improvement, such as figures 1,3,5. 

Response 5: We have increased the resolution of Figure 1,3,5 so that all reviewers can clearly view the images.

3.Response to Comments on the Quality of English Language

Point: The language is more or less correct, but the level could be improved. There are some typos, and grammar errors that need to be corrected.

Response: We have corrected several typos and grammar errors in line 67,561,562-563.

Reviewer 2 Report

Comments and Suggestions for Authors

Comments can be found in the attachment.

Author Response

1.Summary

Thank you very much for taking the time to review this manuscript. Please find the detailed responses below and the corresponding corrections highlighted in the re-submitted files.

2.Point-by-point response to Comments and Suggestions for Authors

Comments 1: In line 67, the sentence structure is incomplete. Please make the necessary corrections. 

Response 1: We have corrected the error in this sentence in line 67.

Comments 2: The figure 1 and figure 9 are a bit blurry. Please replace them with clearer images. 

Response 2: We have increased the resolution of figure 1 and figure 9 so that all reviewers can clearly view the images. It should be noted that we have swapped the order of figure 9 and figure 10.

Comments 3: The title and content of Table 1 do not match. Please make the necessary corrections.

Response 3: We have remade the title of Table 1.

Comments 4: In line 533, the learning rate parameters are missing. Please make the necessary corrections.  

Response 4: We have added the rate parameters in line 561-562.

Comments 5: In line 534, the tense of this sentence is inconsistent with the context. Please make the necessary corrections.

Response 5: We have corrected the error in this sentence in line 562-563.

Comments 6: In Figure 8 (b) and (e), there are errors with the legend. Please make the necessary corrections. 

Response 6: We have corrected the error in Figure 8 (b) and (e).

Comments 7: In line 575.''trackers based on CNN feature extraction and correlation matching show good tracking performance'', but we do not know which methods are based on CNN feature extraction. Please add a column in Table 2 to denote which methods are based on CNN feature extraction and which methods are based on one-stage Transformer.

Response 7: We have added a column in Table 2 to denote which methods are based on CNN feature extraction and which methods are based on one-stage Transformer.

Comments 8: Figure 8, Figure 10, and Table 3 are spread across pages. Please adjust layout. 

Response 8: We have rearranged the article layout so that both Figure 8, Figure 10, and Table 3 are displayed on one or two pages.

Comments 9: Some feature extraction methods summarized in section 2.3 are not recently published, such as, work [42] is published in 2015, and work [43] is published in 2016. More recent works should be cited and discussed. For example, in “Learning discriminative feature representation with pixel-level supervision for forest smoke recognition’, the authors propose a method to extract and fusion local-global features, and in “Ms2net: Multi-scale and multi-stage feature fusion for blurred image super-resolution, the authors propose a method to extract and fusion multi-scale and multi-stage features. please read these papers and discuss them in your work.

Response 9: We have read these two papers and discussed it in our article in line 262-270.

Comments 10: The authors propose a method that combines the local feature extraction capabilities of convolutional neural networks and the global modeling ability of Transformer for object tracking for the first time. This is a good innovation point.

Response 10: We greatly appreciate the reviewer's recognition of this article, and we will continue to optimize it.

Reviewer 3 Report

Comments and Suggestions for Authors

Having studied the work, I would like to note a very accessible and understandable description of the authors' contribution to the application and combination of two machine learning tools (CNN and ViT) to solve an urgent problem.

The methodology of data extraction and the architecture of the models used are presented in detail. The mathematical description and algorithms of data transformation are presented, all this allows us to evaluate the contribution and novelty of the research. I have no serious comments on this section.

The experimental section is also quite detailed, it includes a comparison of the developed approach with alternative solutions on various datasets, as well as checking how the use of a combination of CNN and ViT with bidirectional communication provides better performance than removing any of the components of the approach.

Thus, the article can be accepted in its current form. The only thing that can be corrected in the text is to add brief quantitative results to the annotation, and also in table 1 I did not see the highlighting in different colors.

Author Response

1.Summary

Thank you very much for taking the time to review this manuscript. Please find the detailed responses below and the corresponding corrections highlighted in the re-submitted files.

2.Point-by-point response to Comments and Suggestions for Authors

Comments : Having studied the work, I would like to note a very accessible and understandable description of the authors' contribution to the application and combination of two machine learning tools (CNN and ViT) to solve an urgent problem.

The methodology of data extraction and the architecture of the models used are presented in detail. The mathematical description and algorithms of data transformation are presented, all this allows us to evaluate the contribution and novelty of the research. I have no serious comments on this section.

The experimental section is also quite detailed, it includes a comparison of the developed approach with alternative solutions on various datasets, as well as checking how the use of a combination of CNN and ViT with bidirectional communication provides better performance than removing any of the components of the approach.

Thus, the article can be accepted in its current form. The only thing that can be corrected in the text is to add brief quantitative results to the annotation, and also in table 1 I did not see the highlighting in different colors.

Response : We have remade the title of Table 1. We greatly appreciate the reviewer's recognition of this article, and we will continue to optimize it.

Round 2

Reviewer 1 Report

Comments and Suggestions for Authors

The style of the figure needs a lot of modification and normalization.

Author Response

1.Summary

Thank you very much for taking the time to review this manuscript. Please find the detailed responses below and the corresponding corrections highlighted in the re-submitted files.

2.Point-by-point response to Comments and Suggestions for Authors

Comments: The style of the figure needs a lot of modification and normalization.

Response: We have modified the fill color and clarity of each block in the Transformer branch, CNN branch and Bidirectional Information Interaction Module in Figure 1 to align with the style of other images throughout the text.

We have modified the fill color of CNN block and Transformer encoder block in Figure 2 to match the color of CNN in other images, making it easier for readers to understand.

We have modified the representation of the CNN branch in Figure 3 to make it more distinct from the Transformer branch.

We have modified the number and form of pooling in Figure 6 to better match the description in the article.

3.Additional clarifications

Thank you to the editors and reviewers for their recognition of this article. We will continue to conduct in-depth research in this research field.